# γδ T Are Significantly Impacted by CLL Burden but Only Mildly Influenced by M-MDSCs

**DOI:** 10.3390/cancers17020254

**Published:** 2025-01-14

**Authors:** Michał Zarobkiewicz, Wioleta Kowalska, Agata Szymańska, Natalia Lehman, Bożena Kowalczyk, Waldemar Tomczak, Agnieszka Bojarska-Junak

**Affiliations:** 1Department of Clinical Immunology, Medical University of Lublin, 20-093 Lublin, Poland; wioleta.kowalska@umlub.pl (W.K.); agata.szymanska@umlub.pl (A.S.); lehmannatalia8@gmail.com (N.L.); agnieszka.bojarska-junak@umlub.pl (A.B.-J.); 2Department of Genetics and Microbiology, Maria Curie-Sklodowska University, 20-033 Lublin, Poland; bozena.kowalczyk@mail.umcs.pl; 3Department of Haematooncology and Bone Marrow Transplantation, Medical University of Lublin, 20-080 Lublin, Poland; waldemar.tomczak@umlub.pl

**Keywords:** γδ T, chronic lymphocytic leukaemia (CLL), immunosuppression, M-MDSC, gamma delta T

## Abstract

This study examines how chronic lymphocytic leukaemia (CLL) affects γδ T cells and investigates the role of M-MDSCs in this process. RNA sequencing revealed minor changes in gene expression, while functional tests showed a slight decrease in cytotoxic activity against CLL cells. The findings suggest that M-MDSCs have a limited impact on activated γδ T cells, and other factors likely contribute to immune suppression in CLL.

## 1. Introduction

Chronic lymphocytic leukaemia (CLL) is the most common leukaemia and one of the most prevalent haematological malignancies in adults [1,2]. CLL affects mainly older individuals, usually 60–70 years old, but is also sometimes diagnosed before the age of 30 [1,3]. Despite remarkable progress in CLL treatment, it remains incurable. Thus, patients are usually observed, and therapy is initiated only for those with signs of progression [4]. Modern drugs aim at specific molecular targets: Btk kinases (e.g., ibrutinib or acalbrutinib), BCL-2 (venetoclax), CD20 (e.g., rituximab), or PI3K (e.g., idelalisib) [5]. Those treatment regimens significantly decrease the overall disease burden but do not cure the disease. This can be achieved only by restoring immune function through combined therapies such as drugs (e.g., checkpoint inhibitors) and cellular immunotherapy [6,7]. Such an integrated approach may, on the one hand, lower the risk of acquiring resistance, but it may also overcome already established resistance [8].

γδ T cells form a significant but heterogeneous subset of unconventional T cells [9]. Human γδ T cells are frequently divided into subsets based on the variable fragment of the TCR δ chain they express; thus, Vδ1, Vδ2, Vδ3, Vδ4, and Vδ5 can be distinguished [10,11]. Nevertheless, only the first two populations have received real attention so far; hence, we have little understanding of Vδ3–Vδ5 biology [12]. Vδ1 and Vδ2 cells have high cytotoxic potential and are essential for cancer surveillance [13,14]. Vδ2 cells can be easily expanded 100–1000-fold in vitro for subsequent use in cellular immunotherapy [14,15]. Nevertheless, CLL-derived Vδ2 tends to proliferate poorly due to CLL-related changes that are not yet fully understood [16]. Still, recent clinical trials proved that even allogeneic Vδ2 cells can be safely used for immunotherapy of various cancers [17]. However, Vδ2-mediated therapy is usually highly promising in vitro but fails to deliver similar results in vivo, yielding around ⅕ response rate [18]. This may be related to the highly immunosuppressive environment in cancer tissue and in the peripheral blood of cancer patients. γδ T, including both Vδ1 and Vδ2 subsets derived from healthy volunteers, exhibit significant cytotoxicity against CLL cells [19,20]. However, when γδ T are obtained from CLL patients, the cytotoxicity is severely limited [19].

Thus, a better understanding of both the immunosuppressive environment and γδ T cells in CLL seems necessary for developing successful cellular immunotherapies. The presented study explores the CLL influence on γδ T cells and assesses the impact of M-MDSCs (monocytic myeloid-derived suppressor cells) on γδ T cells in vitro in an attempt to better understand the sources of immunosuppression.

## 2. Materials and Methods

### 2.1. Participants

A total of 163 patients diagnosed with CLL were recruited from the Department of Haemato-oncology and Bone Marrow Transplantation, the Medical University of Lublin, between 03/2016 and 11/2021. All patients were treatment-naive at the time of blood donation. The advancement of the disease was assessed according to Rai’s criteria [21]. The exclusion criteria include other cancer medical history and autoimmune or immune deficiency disease. The control group comprised 34 healthy individuals of similar age and sex composition. Peripheral blood mononuclear cells (PBMCs) were isolated by gradient centrifugation using Gradisol L (Cat No.: 9003.1, Aqua-Med, Łódź, Poland).

The study protocol was approved by the Ethics Committee of the Medical University of Lublin (No. KE-0254/88/2016, KE-0254/176/2020). Written informed consent was obtained from all patients.

### 2.2. γδ. T Stainings

PBMCs were stained with the following anti-human monoclonal antibodies: FITC anti-TCRγδ (BioLegend, San Diego, CA, USA, #331208), PE anti-CD4 (Becton Dickinson [BD], Franklin Lakes, NJ, USA, #555347), PE-Cy5 anti-CD3 (BD, #555341), APC anti-CD8 (BD, #555369), BV510 anti-iNKT (TCR Vα24Jα18; BD, #563267), FITC anti-CD3 (BD, #555332), PE anti-TCRγδ (BioLegend, #331210), and PE-Cy5 anti-CD3 (BD, #555341). The expression of CD4 and CD8 was analysed in 30 patients. Samples were stained for 30 min in darkness at around 4 °C, washed with PBS, and acquired with Cytoflex LX (Beckman Coulter, Brea, CA, USA) and BD FACS Calibur. The gating strategy is presented in Appendix A.

In a subset of patients, TRAIL, NKG2D, Fas, and FasL expression on γδ T cells was also assessed. Similarly, PBMCs were stained for 30 min with the following monoclonal antibodies: FITC anti-TCRγδ (BD, #347903), PE-Cy5 anti-CD3 (BD, #555341), PE anti-TRAIL (R&D Systems, Bio-techne, Minneapolis, MN, USA, #FAB687P), PE anti-NKG2D (BioLegend, #320806), PE anti-Fas (BioLegend, #305608), and PE anti-FasL (BioLegend, #306407). Fluorescent minus one (FMO) controls were used to set the gating.

The M-MDSCs percentage was collected for a subset of the group as part of another project; thus, in the current study, it is only used for correlations and to divide patients into M-MDSClow and M-MDSChigh groups. Complete data for M-MDSCs have already been published [22].

### 2.3. Molecular Biology

γδ T cells from a random selection of 23 patients and eight controls were sorted using BD FACS Aria IIu. First, PBMCs were stained with monoclonal antibodies: FITC anti-TCRγδ (BioLegend, #331208) and PE-Cy5 anti-CD3 (BD, #555341). After washing with PBS, cells were acquired and sorted using BD FACS Aria IIu. The purity of each sort was assessed directly after sorting; only samples with purity >95% were used further.

Sorted cells were stored at −80 °C in RLT buffer (Qiagen, Hilden, Germany #79216) with β-mercaptoethanol. Once all samples were collected, total RNA was isolated using a Blood Mini RNA Isolation kit (Qiagen, #52304), following manufacturer protocol. Then, cDNA was synthesised using Transcriptor First Strand cDNA Synthesis Kit (Roche, Basel, Switzerland, #04379012001). Finally, the expression of cytotoxicity-related mRNA for granzyme A, granzyme B, perforin, ZAP-70, and CD3zeta were assessed with the respective TaqMan Probes: Hs00989184_m1, Hs01554355_m1, Hs00169473_m1, Hs00277148_m1, Hs00609515_m1, all from ThermoFisher Scientific, Waltham, MA, USA. Human GAPDH Endogenous Control (ThermoFisher, #4310884E) was used for normalisation. All reactions were run with TaqMan Universal MasterMix (ThermoFisher, #4304437) on ABI 7300 (Applied Biosciences, ThermoFisher Scientific). Data are presented as 2^−dCT^.

### 2.4. M-MDSC Generation

M-MDSCs were generated as proposed by Okada et al. CD14+ cells were isolated by magnetic-activated cell sorting with anti-CD14 beads (BD, #557769) from healthy volunteer PBMCs [23]. Only those monocytes with purity >90% were used for in vitro generation of M-MDSCs. M-MDSCs were generated by culturing CD14+ cells for 5 days in RPMI-1640 (EuroClone, Pero, Italy, #ECM0495L) + 10% FBS (PanBiotech GmbH, Aidenbach, Germany #P30-19375]) mixed 1:1 with conditioned medium from 786.O kidney cancer cell line. On day 5, CD14^+^ cells were collected, counted, and tested by staining for HLA-DR expression. A significant drop compared to monocytes cultured in the fresh medium was always noted. The 786.O line was obtained from CLS Cell Lines Service GmbH (Eppheim, Germany, #300107), the conditioned medium was obtained by harvesting the culture supernatant at 70–80% confluency of the cell line, and the supernatant was then filtered through 0.22 μm filter to prevent any direct cellular impurities.

### 2.5. Cytotoxicity Tests

γδ T (Vδ2) cells were expanded in vitro following the well-established protocol [24]. Briefly, PBMCs from healthy donors with Vδ2 > 1% of total T cells were seeded into plates at a density of 10^6^ cells/mL in a standard medium (RPMI-1640 [EuroClone, #ECM0495L] + 10% FBS [PanBioTech, #P30-19375]). Zoledronate (Cayman, Ann Arbor, MI, USA, #14984) was added on day 0 at a final concentration 2.5 μM. Cells were fed 100 IU of IL-2 (BioLegend, #589106) on day 0 and every two days. After 2 weeks, the purity was assessed with FITC anti-TCRγδ (BioLegend, #331208) and beFluor647 anti-CD3 (BioEastern, Lublin, Poland, #433164). Cells were used for cytotoxicity only if the purity exceeded 95% on day 14. γδ T cells were incubated ± M-MDSCs for two additional days at a ratio 1:1 (γδ T: M-MDSCs) in 24-well plates with inserts (SPL Life Sciences, Gyeonggi, Republic of Korea, #35324), with both subsets being on two sides of 0.4 μm semi-permeable membrane. Additionally, phospho-vitamin C (Cayman, #16457, conc 100 μg/mL) and/or vitamin D (Cayman, #71820, conc: 10 nM) were added. The co-culture lasted for 48 h. Cytotoxicity was tested on a CLL-derived line Duller, kindly provided by Prof. Dieter Kabelitz (UKSH, Kiel, Germany).

Duller cells were collected, washed with RPMI-1640 + 10% FBS, counted and seeded into 96-well V-bottom plates. γδ T were seeded with Duller cells at a ratio of 10:1 and stimulated with HMBPP (final concentration: 10 nM) (Cayman, #13580). After 4 h at 37 °C, 5% CO_2_ cells were stained with anti-CD3 APC (BioLegend, #300439) and anti-CD107a APC-Cy7 (BioLegend, #328630). Cells were then acquired with Cytoflex LX (Beckman Coulter). Additionally, supernatant was collected and stored at −80 °C for further analysis of soluble mediators. The LEGENDPlex NK/CD8 pre-defined panel (BioLegend, #741187) was used, and the manufacturer manual was followed. This panel allows simultaneous quantification of 13 human proteins, including IL-2, IL-4, IL-6, IL-10, IL-17A, IFN-γ, TNF-α, soluble Fas, soluble FasL, Granzyme A, Granzyme B, Perforin, and Granulysin.

### 2.6. Cytotoxic Synapse Formation and Termination

γδ T cells were pre-exposed to M-MDSCs as described above. γδ T cells were then stained with PKH67 (Sigma-Aldrich/Merck Life Science, Poznań, Poland, #MINI67) and Duller cells with PKH26 (Sigma-Aldrich/Merck Life Science, #MINI26), according to the manufacturer’s protocol. Both γδ T and Duller cells were resuspended in fresh medium (RPMI, 10%FBS) at 10^6^ cells/mL density. Finally, they were mixed and incubated at 37 °C 5% CO_2_. Samples were taken after 0, 20, 40, 60, and 80 min and immediately fixed with paraformaldehyde. Similarly, additional samples were incubated for 60 min to form the cytotoxic synapses and then placed in the thermomixer (37 °C, 1400 rpm). Similarly, samples were taken and immediately fixed after 0, 20, 40, 60, and 80 min. All samples were then acquired with Cytoflex LX, and the percentage of double-stained dimers among all double and single-stained cells was assessed.

### 2.7. Expression of Transcription Factors

To further assess the impact of M-MDSCs on γδ T cells, total RNA was isolated from γδ T co-cultures with M-MDSCs as described above. ReliaPrep RNA Miniprep kits from (Promega Corporation, Madison, WI, USA, #Z6011) were used for RNA isolation. Reverse transcription was performed with a QuantiTect kit (Qiagen, #205311). Expression of mRNA for FoxP3, RORγT, GATA3, and T-bet was assessed through qPCR using GoTaq Probe qPCR MasterMix (Promega, #A6102) and the following TaqMan probes (all from ThermoFisher): ACTB (#4326315E), TBX21 (#Hs00894392_m1), NFIL3 (#Hs00705412_s1), FOXP3 (#Hs01085834_m1), RORC (#Hs01076112_m1), PD1 (#Hs01550088_m1), PDL1 (#Hs00204257_m1). Reactions were run on ABI 7300 (ThermoFisher). Expression was normalised to ACTB using 2^−dCT^.

### 2.8. RNAseq

Total RNA isolated from γδ T cells co-cultured with M-MDSCs as previously described or control γδ T from those same donors was sent for RNAseq to Biomarker Technologies (BMK) GmbH (Münster, Germany). A PE150 sequencing of 6 Gb (20 M PE reads) was performed using DNBSEQ-T7—MGI Tech. Data were analysed using a cloud-based Galaxy system https://usegalaxy.org/ (accessed on 7 January 2025). The data quality was assessed with Fast QC; all samples passed the test. Next, data were mapped using RNAStar, and the expression level was evaluated using featureCounts. Finally, DESeq2 was used to quantify the differential expression between paired samples, and volcano plots were prepared. The 500 top differentially expressed genes (out of 21,948) were analysed in Reactome (reactome.org) for pathway enrichment [25]. Additional Gene Ontology analysis with the creation of a bubble plot for cellular component enrichment was performed with SRPlot https://www.bioinformatics.com.cn/srplot (accessed on 7 January 2025).

### 2.9. In Vitro Expansion

PBMCs from CLL patients were depleted of all CD14^+^ cells (negative control) or M-MDSCs (CD14^+^/HLA-DR^low^) by flow cytometry cell sorting with BD FACS Aria IIu. Next, cultures of PBMCs, PBMCs without monocytes, and PBMCs without M-MDSCs were cultured in RPMI-1640 + 10%FBS and activated with zoledronate. IL-2. Vδ2 count was assessed via flow cytometry after 7 and 14 days—samples were stained with FITC anti-TCRγδ (BioLegend) and APC anti-CD3 (BioLegend, #300439, clone: UCHT1); counting beads were also added (SpheroTech, Lake Forest, IL, USA, #ACBP-150-10).

### 2.10. Statistical Analyses

Data were analysed with GraphPad Prism 9 (GraphPad Software, San Diego, CA, USA). Data distribution was assessed using the Shapiro–Wilk test. The Mann–Whitney U test was used to calculate the *p* values for non-normally distributed data, while the student T-test was used for normally distributed data. Patients were divided into M-MDSC^low^ and M-MDSC^high^ groups according to our previously developed cut-off point (9.35%) [22]. Cytotoxicity data was normalised with z-normalisation due to high variability between different γδ T donors.

The survival analysis was conducted in Statistica 12 (StatSoft Polska, Kraków, Poland). The Kaplan–Meier curves were used to present the overall survival (OS) and time to treatment (TTT). Patients were divided into groups based on γδ T or γδ Tdim percentages. The receiver operating characteristics (ROC) analysis was used to select the cut-off points. The Youden index and AUC (area under the ROC curve) were also calculated. The hazard ratios (HRs) were calculated using Cox proportional hazard models.

## 3. Results

### 3.1. The Distribution of Circulating γδ T into CD4^+^/CD8^+^ and Dim/Bright Subsets Differs Significantly

Initially, we assessed the percentage of total γδ T cells in the peripheral blood of CLL patients. No significant difference was observed (Figure 1A,B).

γδ T lymphocytes express CD4 and CD8. In our cohort, the percentage of γδ T CD8^+^ cells was significantly higher in CLL patients compared to healthy controls, but no difference in the percentage of γδ T CD4^+^ was noted (Figure 1C).

γδ T cells can be divided into dim/bright subsets (Figure 1D,E) based on the CD3 and TCRγδ expression. The dim subset usually dominates in healthy volunteers; the opposite trend was observed in CLL patients. The differences in both dim and bright subsets were significant (Figure 1D). Next, the expression of Fas, FasL, NKG2D, and TRAIL was assessed in a subset of patients. FasL and TRAIL were significantly downregulated on γδ T cells from CLL patients, while no differences were noted for Fas and NKG2D (Figure 1F).

To assess the potential impact of γδ T cells on the course of CLL and vice versa, we analysed the percentage of γδ T lymphocytes according to the stage of disease and compared patients with and without unfavourable prognostic factors. The dim γδ T subset dominated the bright one in CD38 negative patients (Figure 1G), while it did not differ between ZAP-70^+^ and ZAP-70^−^ ones. Nevertheless, ZAP-70^+^ patients had a significantly lower percentage of total γδ T (Figure 1H). Similarly to CD38 negative patients, those with mutated IGHV showed domination of dim over bright subset (Figure 1I). There were no differences when patients were divided based on Rai stage, cytogenetic abnormalities, survival, or treatment requirement (Appendix A).

### 3.2. There Is No Relation Between γδ T Cells and Overall Survival

Next, we analysed the impact of γδ T cells and their dim/bright subsets on overall survival and time-to-treatment of CLL patients. Patients were divided into low and high γδ T (or dim γδ T) percentages based on ROC analysis. There were no significant differences in overall survival or time-to-treatment for low and high groups (Figure 2).

### 3.3. Higher ZAP-70 Expression in γδ T Cells from CLL Patients

Next, we analysed the expression of selected mRNAs in isolated γδ T cells. Initially, the downregulation of FasL was confirmed (Figure 3A). Next, we looked over the expressions of other cytotoxicity-related molecules—perforin, granzyme B, GPR65, and TRAIL with no significant differences (Figure 3B–E). Finally, the mRNA expression of CD3 zeta and ZAP-70, two important TCR-signalling molecules, was assessed; there was no difference in CD3 zeta (Figure 3F), while ZAP-70 was significantly overexpressed in CLL-derived γδ T cells (Figure 3G).

### 3.4. M-MDSCs Do Not Significantly Affect the Cytotoxic Potential of γδ T Cells

γδ T cells were first pre-incubated for 48 h with M-MDSCs at a ratio of 1:1 (γδ T: M-MDSC) and then used as effector cells against CLL-derived line Duller [26]. An insignificant but noticeable drop in cytotoxicity of γδ T was observed (Figure 4A). Additionally, phospho-vitamin C was tested as a potential positive regulator of γδ T cytotoxicity along with vitamin D, a potential negative regulator of M-MDSC function [27,28]. Neither of those vitamins impacted the cytotoxicity. M-MDSC-exposed γδ T cells expressed increased levels of IL-10 (Figure 4B). While exposure to M-MDSCs did not affect the expression of PD-1 or its ligand, it significantly decreased that of *RORC* (Figure 4C). To fully comprehend the impact of M-MDSC on γδ T cytotoxic potential, we analysed the formation and stability of γδ T: Duller cytotoxic synapse. M-MDSC exposure induced insignificant changes in synapse formation and no changes in synapse stability (Appendix A).

### 3.5. Depletion of M-MDSCs Does Not Rescue Vδ2 Reactivity to Phosphoantigens

γδ T (Vδ2) can be easily expanded from PBMCs by stimulation with either aminobisphosphonate + IL-2 or phosphoantigen (e.g., IPP) + IL-2. CLL-derived γδ T were previously reported to be not responsive [19]. M-MDSC-depletion did not increase the proliferation of γδ T cells in zoledronate-stimulated PBMCs from CLL patients (Figure 4D).

### 3.6. M-MDSCs Significantly Regulate the Response to Stimulation and Survival of γδ T Cells

Exposure to M-MDSCs led to a significant change in the transcriptome of γδ T cells (Figure 5A). Pathway enrichment analysis revealed two substantial groups of differentially regulated genes related to cellular stimulation and transcription of p53-related genes (Figure 5B). Cellular response to hypoxia and oxidative stress were among the sub-pathways (Full-size diagrams: Appendix A). Interestingly, *BAX* and *PHLDA3* are among the top 50 differentially regulated genes with a significantly lower expression in MDSCs-exposed γδ T cells. On the other hand, *NUDC*, a gene critical for mitosis, is also significantly downregulated.

## 4. Discussion

CLL burden significantly impacts overall immune response. Human γδ T functions in a vast crosstalk network with other immune cells. Some of those interactions result in activation and better anti-tumour or anti-viral response. In contrast, others, most notably with neutrophils and MDSCs, result in at least partial suppression of γδ T cells [29]. As previously demonstrated, M-MDSCs are a vital part of the immunosuppressive landscape in CLL [22]. Indeed, PMN-MDSCs (polymorphonuclear MDSCs) infiltration of tumours correlates with a lower cytotoxic potential of tumour-infiltrating T cells [30]. Moreover, MDSCs significantly hamper the efficacy of novel treatment modalities, including checkpoint inhibitors [31].

Interestingly, exposure to M-MDSCs at the time of cytotoxic T-cell expansion leads to cells that remain undifferentiated. Still, once those cells are transferred to a tumour-bearing animal host, they show superior cytotoxicity compared to cytotoxic T cells that were not exposed [32]. Alessandra Sacchi et al. reported a significant suppression of Vδ2 γδ T cells exposed to PMN-MDSCs. Interestingly, they noted impaired degranulation, lower IFN-γ production and decreased cytotoxic response against Daudi and Jurkat, two cell lines potently killed by human γδ T cells [33]. This is in striking contrast to our results, in which only a minor impact of M-MDSCs was noted. This suggests that human γδ T cells are more susceptible to PMN-MDSC suppression than M-MDSC. Still, PMN-MDSCs usually dominate over M-MDSCs in human cancer patients [34]. Ferrer et al. noticed that PMN-MDSCs are more potent immunosuppressors than M-MDSCs in CLL patients [35].

MDSCs, apart from directly suppressing T cells, induce the generation of T regulatory cells [36,37]. Traditionally, T regulatory cells are associated with αβ lineage, but as demonstrated by Peters et al., Treg-like cells can also be generated from γδ T ones [38]. Although we have not observed any significant upregulation of FoxP3, we noted a significant increase in the production of IL-10 by γδ T exposed to M-MDSCs. IL-10, on the one hand, promotes further accumulation of MDSCs, but on the other, it hampers the IFN-γ production and cytotoxic potential of T cells [36]. The limited effect of co-cultures should probably be attributed to the limited co-culture time. In vitro expanded γδ T cells have a restricted lifespan; thus, long co-cultures that could better resemble in vivo settings seem impossible. Moreover, our co-culture model restricted direct cell–cell interaction and the suppression was mostly conveyed via soluble factors, while normally, MDSCs also partially induced suppression via direct interactions [36].

Human Vδ2 cells are highly suppressed and dysfunctional in CLL patients [39]. They proliferate poorly in response to phosphoantigen stimulation; the extent of response seems to have a limited prognostic value in CLL [16]. We hypothesised that this may be MDSC-driven, but the depletion of M-MDSCs did not improve the proliferation. To mimic the CLL microenvironment, we have generated M-MDSCs in vitro from healthy donor monocytes and co-cultured them with in vitro expanded human γδ T cells. We have observed a noticeable decrease in cytotoxic potential and higher IL-10 production. To better understand this phenomenon, we performed RNAseq analysis and discovered significant changes in the transcriptome of γδ T cells exposed to M-MDSCs. MDSCs caused the downregulation of various genes crucial for the proliferation and activation of γδ T cells. MDSCs lowered the expression of *BAX*, an important pro-apoptotic factor [40]. This could partially explain the accumulation of anergic γδ T cells in CLL patients. Interestingly, co-cultured γδ T cells also had lower *PHLDA3* and *NUDC*. While the exact role of the former gene in lymphocytes remains unclear, it is upregulated in fast-proliferating cancers, suggesting its role in proliferation [41,42]. Finally, the latter is crucial for cytokinesis during mitosis [43]. Altogether, this indicates that M-MDSCs may, on the one hand, hamper γδ T proliferation but, on the other hand, prolong their survival by decreasing the apoptosis.

Moreover, MDSCs significantly regulated pathways of cellular response to stimuli and stress, including those reacting to hypoxia and oxidative stress. Hypoxia is one of the core problems of cancer biology, as it hampers not only the natural response of cytotoxic immune cells but also decreases the efficacy of immunotherapy [44,45,46]. Interestingly, tumour hypoxia regulates γδ T cytotoxic potential directly and indirectly via MDSCs as it promotes MDSCs immunosuppression via miR-21/PD-L1 axis [47]. Additionally, tumour hypoxia *per se* promotes IL-17 production and PD-L1 expression by γδ T cells, thus promoting anti-inflammatory phenotype [48]. Altogether, it suggests that immunomodulation of MDSC function may facilitate a better adaptability to hypoxia and improve cytotoxic response against cancer.

Contrary to the previous report of Bartkowiak et al. [49], we have not observed any significant accumulation of γδ T cells in CLL patients. Still, we have observed a higher expression of TCR on γδ T from CLL patients (dim vs. bright). This suggests lower activation. This aligns with Coscia and de Weerdt, who reported decreased cytotoxic potential and lower granzyme B, TNF, and IFN-γ expression in CLL-derived γδ T cells [16,19,50]. Similarly to Własiuk et al. and Bartkowiak et al., we have not noticed any significant correlation between peripheral γδ T cells and the clinical course of CLL [49,51].

Nevertheless, due to the limited material availability, the current study utilised in vitro-generated M-MDSCs instead of M-MDSCs isolated directly from patients’ blood. Such generated M-MDSCs are widely used and accepted in research but usually are not fully equal to those derived from cancer patient blood [52]. Similarly, the in vitro expanded γδ T cells do not confer the full heterogeneity that could be attributed to a total pool of γδ T circulating the peripheral blood. Still, zoledronate-generated γδ T cells form the basis of γδ T research worldwide. Finally, the Duller cell line used in this project was originally classified as centrocytic B-cell lymphoma, based on the now obsolete Kiel criteria for diagnosing hematological malignancies. However, both the patient’s clinical data and the immunophenotype of Duller cells meet the modern diagnostic criteria for CLL [26,53]. Duller cells exhibit a typical CLL phenotype: CD19^+^/CD5^+^/CD23^+^ with surface immunoglobulin expression and restriction to a single light chain (lambda). In contrast, the widely used MEC-1 and MEC-2 cell lines lack CD5 expression, with MEC-2 also being CD23-negative [54].

## 5. Conclusions

In vitro expanded human γδ T cells retain most of their function after short-time exposure to M-MDSCs. Ex vivo data suggest that long-term exposure to a highly immunosuppressive environment of CLL renders most γδ T anergic. Altogether, from the immunotherapeutic point of view, the optimal treatment regimen with γδ T cells would include multiple doses, with just a few days between the infusions.

## Figures and Tables

**Figure 1 cancers-17-00254-f001:**
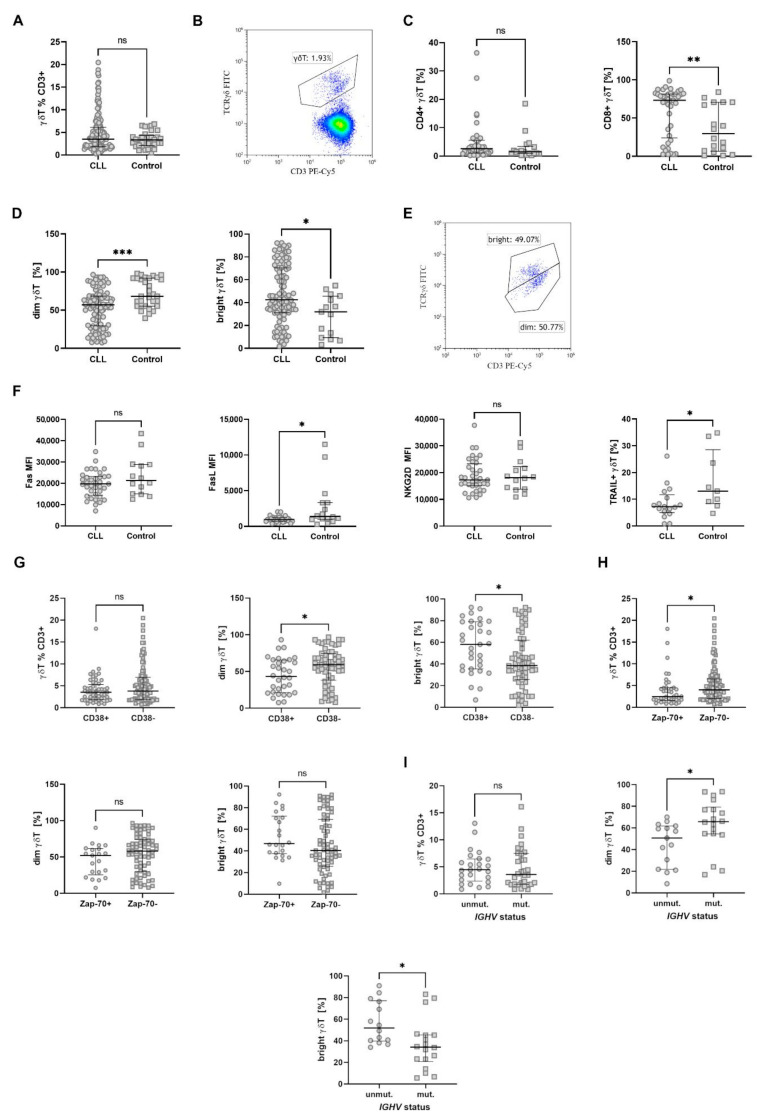
Results of basic immunophenotyping. Panel (**A**) shows the overall percentage of γδ T cells among all T lymphocytes (CLL *n* = 166, control *n* = 33). Panel (**B**) illustrates an example of the gating strategy used. Panel (**C**) displays CD4 and CD8 expressions on γδ T cells (CLL *n* = 41, control *n* = 18). Panel (**D**) shows the distribution of dim and bright subsets of γδ T cells (CLL *n* = 104, control *n* = 15), with a gating example shown in Panel (**E**). Panel (**F**) presents the expression of Fas (CLL *n* = 37, control *n* = 14), FasL (CLL *n* = 34, control *n* = 16), NKG2D (CLL *n* = 36, control *n* = 14), and TRAIL (CLL *n* = 17, control *n* = 9) on γδ T cells. Panels (**G**–**I**) depict γδ T cells stratified by disease stage and major prognostic factors (CD38, Zap-70 expression, *IGHV* mutations; patient numbers correspond to those in Panels (**A**–**F**). * *p* < 0.05, ** *p* < 0.01, *** *p* < 0.001, ns, not significant; MFI, mean fluorescence intensity.

**Figure 2 cancers-17-00254-f002:**
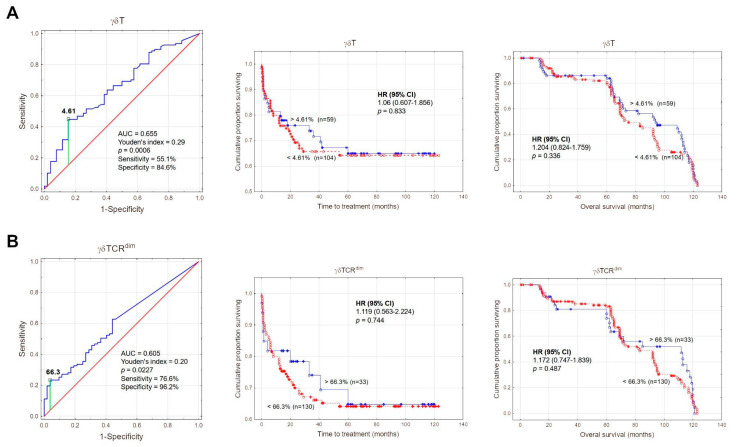
Clinical importance of γδ T cells in CLL. ROC analysis was performed to determine the optimal cut-off values for γδ T cell percentages (**A**) and γδ T^dim^ cell percentages (**B**) that best distinguished ZAP-70-positive from ZAP-70-negative cases, using AUC and the Youden index. Kaplan–Meier curves were used to evaluate time-to-treatment (TTT) and overall survival (OS). Panel (**A**): A cut-off of 4.61% for γδ T cell percentage was selected, corresponding to a sensitivity of 55.1% and specificity of 84.6%. The blue line represents patients with γδ T cell levels > 4.61% (n = 59), while the red line represents those with levels ≤ 4.61% (n = 104). No significant difference was observed for TTT (HR = 1.06, *p* = 0.833) or OS (HR = 1.204, *p* = 0.336). Panel (**B**): A cut-off of 66.3% for γδ T^dim^ cell percentage was chosen, with a sensitivity of 76.6% and specificity of 96.2%. The blue line represents patients with γδ T^dim^ levels > 66.3% (n = 33), while the red line represents those with levels ≤ 66.3% (n = 130). No significant difference was observed for TTT (HR = 1.119, *p* = 0.744) or OS (HR = 1.172, *p* = 0.487).

**Figure 3 cancers-17-00254-f003:**
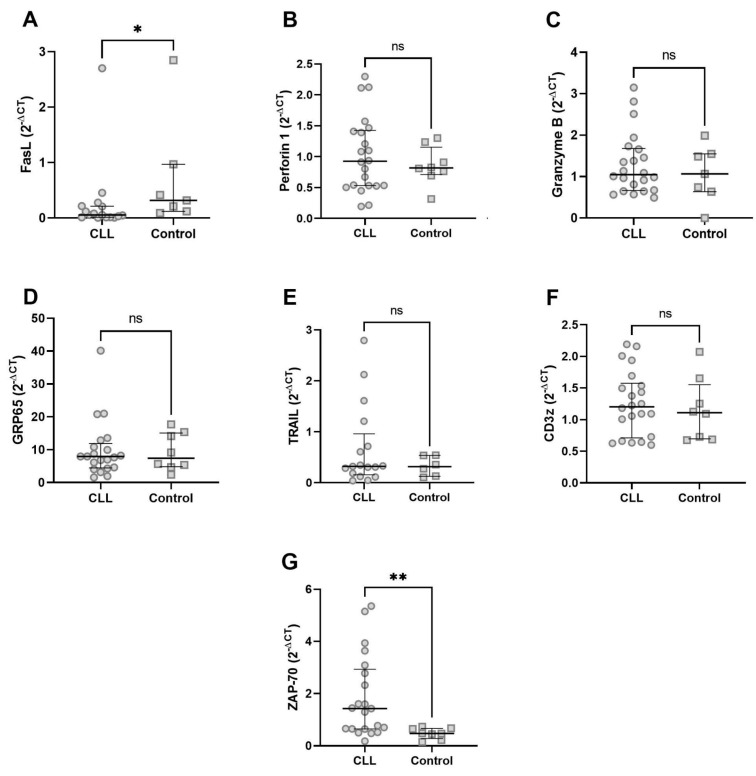
Expression of selected mRNAs in γδ T cells from CLL patients [CLL n = 16–22, control n = 6–8]—*FASL* (**A**), *PERF1* (**B**), *GZB* (**C**), *GPR65* (**D**), *TRAIL* (**E**), *CD247* (**F**), *ZAP70* (**G**). * *p* < 0.05, ** *p* < 0.01, ns, no significance.

**Figure 4 cancers-17-00254-f004:**
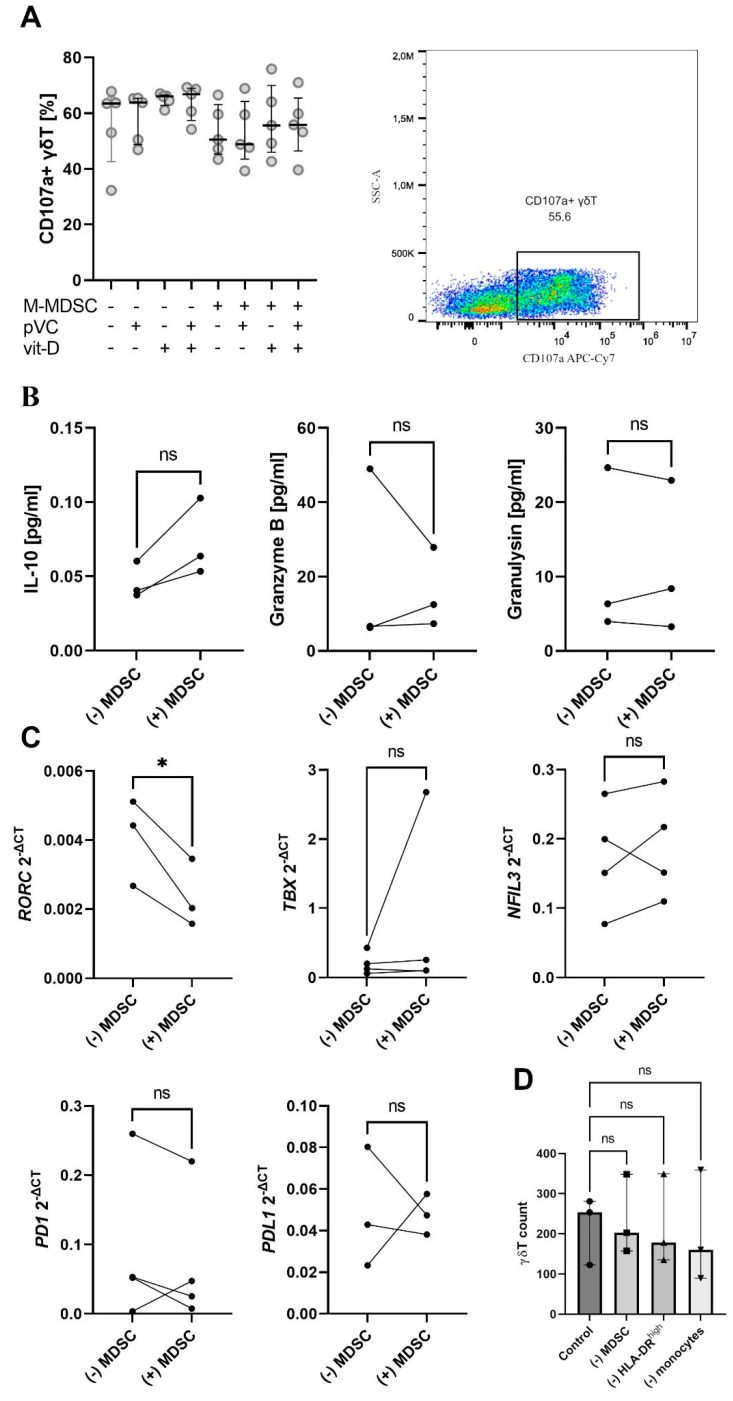
Impact of M-MDSCs on γδ T cytotoxicity against CLL cells [Duller], as well as γδ T cells or γδ T: M-MDSC co-cultures were preincubated with vitamin C ± vitamin D for 48 h. Cytotoxicity was assessed indirectly by measuring CD107a expression; the dot-plot shows exemplary staining for the M-MDSC co-cultured sample with preincubation with both vitamins (**A**). The experiment was repeated five times. The concentration of IL-10, granzyme B and granulysin were measured with LegendPlex (**B**), and the expression of various mRNAs was assessed with qPCR (**C**). The impact of M-MDSCs on CLL-derived γδ T proliferation was negligible—almost no proliferation was observed when total CLL-derived PBMCs were cultured, or when PBMCs with either of the depleted subsets (M-MDSCs, HLA-DRhi monocytes, total monocytes) were used (**D**). * *p* < 0.05, ns, no significance, pVC—phospho-vitamin C, vitD—vitamin D.

**Figure 5 cancers-17-00254-f005:**
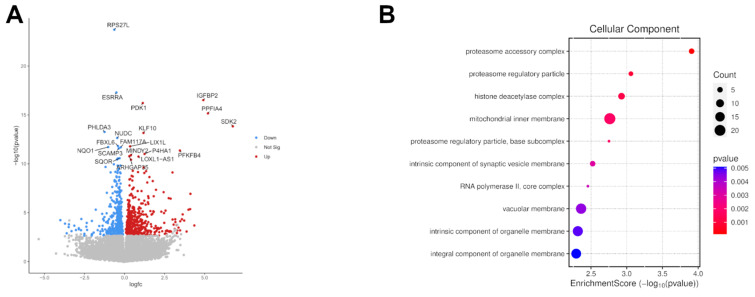
Transcriptomic profiling and functional enrichment analysis of differentially expressed genes (DEGs). (**A**) Volcano plot displays a change in expression level versus *p*-value; significantly upregulated genes are marked red, while downregulated ones are blue. The top 20 genes are marked by their symbols. (**B**) Bubble plot showing Gene Ontology (GO) analysis, showing the most enriched cellular components. The dot size represents the count of associated genes, while the colour gradient indicates the *p*-value.

## Data Availability

The data generated and analyzed during this study are available from the corresponding author upon reasonable request.

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
