# Peer review of "γδ T Are Significantly Impacted by CLL Burden but Only Mildly Influenced by M-MDSCs"

_cancers, 2025, doi:10.3390/cancers17020254_

Round 1

Reviewer 1 Report

Comments and Suggestions for Authors

Present article by Zarobkiewicz et al “γδ T are significantly impacted by CLL burden, but only mildly influenced by M-MDSCs” is nicely presented work. I have following comments:

1. Introduction line 36-37 “CLL affects mostly older individuals, usually 60-70 years old, but is also sometimes diagnosed in 30+ year-olds” seems to exclude patients <30-year-old. However, CLL can also be diagnosed in <30 years also https://pubmed.ncbi.nlm.nih.gov/27463503/ . Kindly make required changes.

2. Line 40- Modern drugs aim at certain molecular targets e.g. Btk kinases this line is incomplete as CLL treatment includes multiple line of therapies.

3. Line 42-42 This can potentially be only ever achieved by restoration of immune function e.g. with combined therapies - drug + cell transfer. This line is not very clear. Kindly rephrase.

4. Line 70 Haematooncology should be separated by a hyphen.

5. Line 87-89 formatting is off

6. Were the CLL patients tested for the presence of deletion 13, Del11, Deletion 17, Trisomy 12 regions. These are the common cytogenetic abnormalities and can influence the treatment outcome.

Author Response

Dear Reviewer, thank you for your kind review. Please see the attachment for the full response.

Reviewer 2 Report

Comments and Suggestions for Authors

The authors present an analysis of the interactions between chronic lymphocytic leukemia (CLL) and γδ T cells, focusing on the potential immunosuppressive role of M-MDSCs. My main concern is the use of a non-typical CLL cell line for the experiments and needs to be clarified. See point 1 of the main comments.

Major comments:

1. The authors evaluate the cytotoxicity of γδ T cells after co-culture with the "CLL-derived line Duller" provided by Prof. Dieter Kabelitz. Please can the authors explain why this cell line derived from a centrocytic B-cell lymphoma was used instead of a more common cell line such as MEC-1?

2. I suggest that the authors clarify section "3.1. The percentage of circulating γδ T does not differ between healthy controls and CLL patients" or merge with the next section as it remains poorly developed. In addition to this point, could the authors please revise the "Figure 1" footnote on line 254.

3. Could the authors explain why they chose these genes to be analyzed by qPCR and not others and how they chose them?

4. Could the authors explain why they chose a 1:1 ratio (γδ T cells: M-MDSCs) and did not show an experiment combining different ratios to elucidate the best one?

5. The authors claim in section "3.5. M-MDSCs negatively affect the cytotoxic potential of γδ T cells" that MDSCs negatively affect the cytotoxicity of γδ T cells, but "Figure 4" and the whole section did not show any evidence of this, actually all data are "ns". I recommend to the authors or increase the "n" or rewrite the title of the section, because with the data provided, no effect of MDSCs on γδ T cells can be elucidated.

6. In section "3.7. M-MDSCs significantly regulate the response to stimulation and survival of γδ T cells" related to RNA-seq data, I suggest the authors to change the way of presenting the altered pathways as the figure is too small, they could use a bubble plot for example. Also, the authors can contextualize the genes they found altered in the analysis (e.g. PHLDA3).

Minor Comments:

1. The authors present very well the disease and its current states, but I would like to suggest that besides talking about BKT inhibitors (i.e. Ibrutinib) they should also mention the other small molecule treatment in CLL: BCL2 inhibitors (i.e. Venetoclax). As the later found differences in the BAX gene.

2. In section “2.2 γδ T staining" the authors repeated the same flow cytometry antibody "PE-Cy5 anti-CD3 (BD, #555341)" in line 84 and 86, I suggest to eliminate the antibody of line 86.

3. In line 135, I recommend to specify the density seeding more correctly. Instead of "106/ml" you should write: "106 cells/ml".

4. In section "2.9. In vitro expansion" the authors use zoledronate + IL-2 for T cell expansion. Could the authors provide a reference where they chose this combination and not the use of IL-18 instead (10.4049/jimmunol.1300603)?

5. In "Supplementary Fig. 1. γδ T gating strategy" I suggest the authors to review the gating of singlets as they select some doublets. They need to fit the diagonal more accurately.

6. I suggest the authors to add the FMO in "Figure 4A" to clarify the cut-off of positivity for CD107a.

7. The authors inform about a "Video S1" that was not included in the submission.

Comments on the Quality of English Language

The authors should carefully revise the English of the manuscript, as some grammatical errors have been found: lines 42-43; 126-127; 138 (every two days, not second days); 292 (instead of lowered, decreased); 392 and 397 (avoid contractions).

Author Response

(The authors gave the same response as above.)

Reviewer 3 Report

Comments and Suggestions for Authors

Thank you for providing an opportunity to review the manuscript titled "γδ T are significantly impacted by CLL burden, but only mildly influenced by M-MDSCs" by MichaÅ‚ Zarobkiewicz et al.

In this manuscript, MichaÅ‚ Zarobkiewicz and co-authors shows how CLL cells and M-MDSCs can impact the γδ T cells and its function. They have used various in vitro co-cultures of γδ T cells with M-MDSCs to understand the immunosuppressive impact of M-MDSCs on γδ T. Altogether, they found that M-MDSCs have limited effect on γδ T cells. 

Major comments:

1. The authors have shown that the CLL patients have higher TCRγδ expression, compared to control. Does the γδ T dim/bright subsets differ in their gene expression profile and functionality? The authors many do not need to prove this through any experimentation. However, the reviewer is curious, and the authors can discuss about it.

2. How does the CD107a levels (Figure 4A) is a measure of cytotoxicity activity of γδ T cells?

Minor comments:

1. Please indicate what red and blue lines represents in Figure 2.

2. The results for the cytotoxic synapse formation were given as supp. fig. 2 in lane 296. This is in fact supplementary figure 3. Please correct it.

Author Response

(The authors gave the same response as above.)

Round 2

Reviewer 2 Report

Comments and Suggestions for Authors

The authors have addressed all comments in a thorough and methodical manner, thereby enhancing the quality of the manuscript. I would like to express my gratitude for their efforts in implementing the changes suggested by the reviewers and for the thorough explanation provided in the rebuttal letter.

However, the clinical data and immunophenotype of the Duller cell line, as referenced in Comment 1, remain to be elucidated. Therefore, the authors are requested to provide these essential elements for a comprehensive evaluation.

Author Response

Dear Reviewer,

Thank you for your feedback on our manuscript. We appreciate your careful review and are pleased to address the final minor comment you provided. Below, we outline our response and the corresponding revision.

We have added the following paragraph to the limitations section of the discussion:

"Finally, the Duller cell line used in this project was originally classified as centrocytic B-cell lymphoma, based on the now obsolete Kiel criteria for diagnosing hematological malignancies. However, both the patient’s clinical data and the immunophenotype of Duller cells meet the modern diagnostic criteria for  CLL [Jenssen et al. 1990, Mulligan et al. 2012]. Duller cells exhibit a typical CLL phenotype: CD19+/CD5+/CD23+ with surface immunoglobulin expression and restriction to a single light chain (lambda). In contrast, the widely used MEC-1 and MEC-2 cell lines lack CD5 expression, with MEC-2 also being CD23-negative  [Stacchini et al. 1999]."

Change in the limitations as well as two new references (53, 54) are marked in yellow.